# The Effect of Working Mother Status on Children's Education Attainment: Evidence from Longitudinal Data

**Siti Nur Azizah** [1,*] **, Samsubar Saleh** [2] **and Eny Sulistyaningrum** [2]

1    Faculty of Islamic Economics and Business, Sekolah Pascasarjana Universitas Gadjah Mada,
     Universitas Islam Negeri Sunan Kalijaga, Yogyakarta 55281, Indonesia
2    Fakultas Ekonomika dan Bisnis, Universitas Gadjah Mada, Yogyakarta 55223, Indonesia;
     samsubar@ugm.ac.id (S.S.); eny@ugm.ac.id (E.S.)
*    Correspondence: siti.azizah@uin-suka.ac.id

**Abstract:** This study builds on the debate on whether mothers' employment in the early life of children bring positive or negative impact to children in later life. This is based on the fact that employment often result in reduced family time in which mother may not be able to take care of their children fully. This study estimated the effects of mothers' employment status on children's education in the short-term and in the long-term in 24 provinces in Indonesia. This study used data on children from three waves of the Indonesia Family Life Survey (IFLS): IFLS-3 (in 2000) aged 0–7 years, IFLS-4 (in 2007) aged 7–14 years, and IFLS-5 (in 2014) aged 14–21 years. The outcome variable was the children's education and the variable of interest was the mother's employment status. To overcome endogeneity, the analysis of the relationship between outcome and interest variables in this study was done by using the ordinary least square estimation (OLS) method and instrumental variables (IV). This study also used a cross-sectional design which estimated IFLS-4 (in 2007) and IFLS-5 (in 2014), separately. The main finding suggests that the mother's employment status positively influenced children's education both in the short- and long-term, as evidenced by the ordinary least square estimation (OLS) results. The instrumental variable (IV) estimate showed that decision-making was a strong instrument; and, it further revealed that mothers' decision-making within the family affected their opportunity to work. This research could contribute to strengthening working mothers' self-border and the concept of work-family facilitation in a family. It could also become a reference for stakeholders involved in the policy making to regulate policies which facilitate and support working mother to create ideal working environment. This condition is expected to supports children's development as well as creating equity for working mothers in Indonesia.

**Keywords:** children's education; working mother status; IFLS; OLS; IV

## 1. Introduction

Education plays a pivotal role in producing quality human resources (Hagglund and Samuelson 2009). With educators, it aims to guide children in the process of dialogic transformation so they can grow and shape their awareness, attitude, and critical thinking (Nurlaela 2013). For this reason, the success of children's education can influence the sustainability and the growth of a nation (Maunah 2019). This is because educational success may create younger generations who possess stronger intellectual capacity and character (Onyekuru and Ibegbunam 2013). This, in turn, could become the driving force for technological advancement and cultural heritage preservation.

According to the Program for International Student Assessment (PISA) survey, Indonesia ranked in the bottom six (TribunNews 2021). One of the survey indicators suggested that the low level of literacy among Indonesian students was due to a feudalistic education system. In response to this poor performance in PISA, the Ministry of Education and Culture launched the concept of "Merdeka Belajar" (Learning Freedom) (Gatra 2021). This

is an educational development concept in which families, teachers, educational institutions, the industrial world, and the community synergize to become agents of change. The three indicators proposed are equal participation of students in Indonesian education, effective learning, and no child left behind (Kemendikbud 2020). The success of this program requires synergy between the school and the parents. In particular, mothers are the main facilitators of children's education (Bunujevac and Durisic 2017; Christenson et al. 2012). Mothers play an important role in the process of children's education as they are capable of molding children's attitude toward education, which affects the first 9 years of schooling (Yulianti et al. 2019); supporting children's achievement in school; and increasing children's test scores (Dunifon et al. 2013; Sapungan and Sapungan 2014).

Several studies have investigated a number of variables which may shape children's educational success including the effect of gender differences (Calvin et al. 2010; Fischer et al. 2013; Huang 2012; Spinath et al. 2014), teacher education (Bunujevac and Durisic 2017; Farooq et al. 2011; Huang 2012; Ramanlingam and Maniam 2020), teaching approaches used (Derbali et al. 2019; Slade and Griffith 2013), school and classroom environments (Gandhi 2017; Harinarayanan and Panzhanivelu 2018; Kaur 2020; Kurniawan et al. 2018; Lawrence and Vimala 2012; Obaki 2017), socioeconomic conditions of students' families (Bhat et al. 2016; Faraz and Noor 2019; Ghaemi and Yazdanpanah 2014; Gobena 2018; Zhan and Sherraden 2003), and the educational background of students' families (Dettmers et al. 2019; Jaiswal 2018; Ribeiro et al. 2021). The research findings also show inconclusive yet interesting findings and this is primarily due to the geographical differences where the research was conducted.

The present research is different from previous research in that it examines the effect of mothers' employment status on children's educational attainment in the short- and long-term. This research was conducted in Indonesia where most of the female population participates in the labor market (Central Bureau of Statistics 2019). Married women with an age range of 17–70 years have decided to work in the formal sector. In 2019, 55.50% of women worked and this figure rose by 0.06% compared to 2018. There was also an increase of 5.50% when compared to the data in 2013 (Central Bureau of Statistics 2019).

The increasing trend of mothers' participation in the labor market has caused major changes in child-rearing (Bettinger et al. 2014), opening up a debate between the positive and negative effects of this trend (Purmini et al. 2016). Some studies suggest that the dual roles of mothers cause conflict in the family which eventually disrupts family harmony. (Anafarta 2011). Presser (2000) suggests that marriages are six times more likely to fail when mothers work. This is because working mothers have little time allocated to interact with children and the family (Singh 2018a). Additionally, Poortman (2005) finds that maternal participation in the workforce is often seen as a cause of increasing divorce rates. Several previous studies have found that a wife's level of income and financial independence affects the quality of a marriage (Schoen et al. 2002). Mothers' level of income increases their self-confidence and support self-actualization (Verma and Negi 2020). It will also increase their decision-making power in the household including decision to divorce (Asghari et al. 2013), (Brinig and Allen 2013; Vignoli et al. 2016; Killewald 2016; Ascandra et al. 2019).

Depending on the family's economic capability, mothers may to assign child care to grandmothers, relatives, assistants, householders, babysitters, and daycare facilities (Ingstad and Hedlund 2017; Samman et al. 2016; Veramendi and Urzua 2011). This has implications for the quality of children's growth (Rizky et al. 2017; Kusumawardhani and Warda 2013; Rahmani et al. 2018). Children whose mothers work tend to lack discipline and self-confidence and they also experience health problems (Lei et al. 2018), including impaired cognitive growth (Stephiana and Wisana 2019). This results in a disruption of the educational process (LaRocque et al. 2011), the emergence of negative behaviors, a decrease in test and exam scores at school (Lei et al. 2018), and decreased academic and non-academic achievements (Dunifon et al. 2013). These negative impacts are strong predictors for children wellbeing in the future (Rozental et al. 2018; McMullen et al. 2020).

On the other hand, apart from wanting to meet economic needs, a strong reason for mothers to participate in the labor market is to provide for children's educational costs (Arifin 2017; OECD 2021; Purmini et al. 2016; Singh 2018b; Verma and Negi 2020). The cost of a child's education is a crucial issue for parents (Poduval and Poduval 2009). The inability to pay educational costs results in a high dropout rate. In this case, family income and wealth support the probability of children's educational success. In addition, the availability of financial supports provided by working mothers may increase a child's chance to attend higher education (i.e., university entrance), which has an impact on their future employment and income.

Therefore, this study tries to determine the extent to which the employment status of mothers affects children's education in the short- and long-term. This research adopts the model of Heckman et al. (2006) and Ruhm (2000) which assume that education is the result of human capital investment, consisting of the accumulation of capital, goods, and time that affect children's outcomes in adulthood. This study also follows the Kingdon and Teal (2002) model which examines the effect of salary and student achievement. Furthermore, this study adopts the work of Hoque et al. (2017) which examines the effect of the mother's profession on the child's academic achievement. It also follows the model developed by Sofa (2019), which investigates the effect of a mother's decision to work on a child's future outcomes. This study uses mother's employment status as an independent variable and the child's education as a dependent variable, which is measured by the length of the child's schooling. The measurement of children's school years was inspired by the research of Purmini et al. (2016).

## 2. Theoretical Background

### 2.1. Education System in Indonesia

The Indonesian education system is regulated by the Law on the National Education System (No. 20/2003). The management of education in Indonesia is regulated by the Ministry of Education and Culture and the Ministry of Religious Affairs. The education system in Indonesia is further divided into three main types, namely formal, non-formal, and informal education (Hendajany et al. 2016). Formal education is education held in schools while non-formal and informal education occur outside the school system. In formal education, there are three levels: basic education (primary), intermediate education (secondary), and higher education (tertiary). Primary education consists of elementary school or madrasah ibtidaiah and junior high school or madrasah tsanawiyah. Secondary education consists of senior high school (madrasah aliyah) and vocational school (SMK). Figure 1 shows the illustration of the education system in Indonesia in accordance with Law No. 20 of 2003. The chart on the left shows the average age of individuals consistent with school level. The middle chart shows the level of education in Indonesia and the rightmost chart shows the types of formal schools in Indonesia.

Informal education consists of courses, internships, Paket A, Paket B, Paket C, playgroups, and childcare. The institution in charge of courses, internships, Paket A, Paket B, and Paket C is the Ministry of National Education, while playgroups and nurseries are under the supervision of the Ministry of National Education and the Ministry of Social Affairs. Figure 1 illustrates the duration of schooling (number of years of schooling), level of education, and type of education as per Law No. 20/2003.

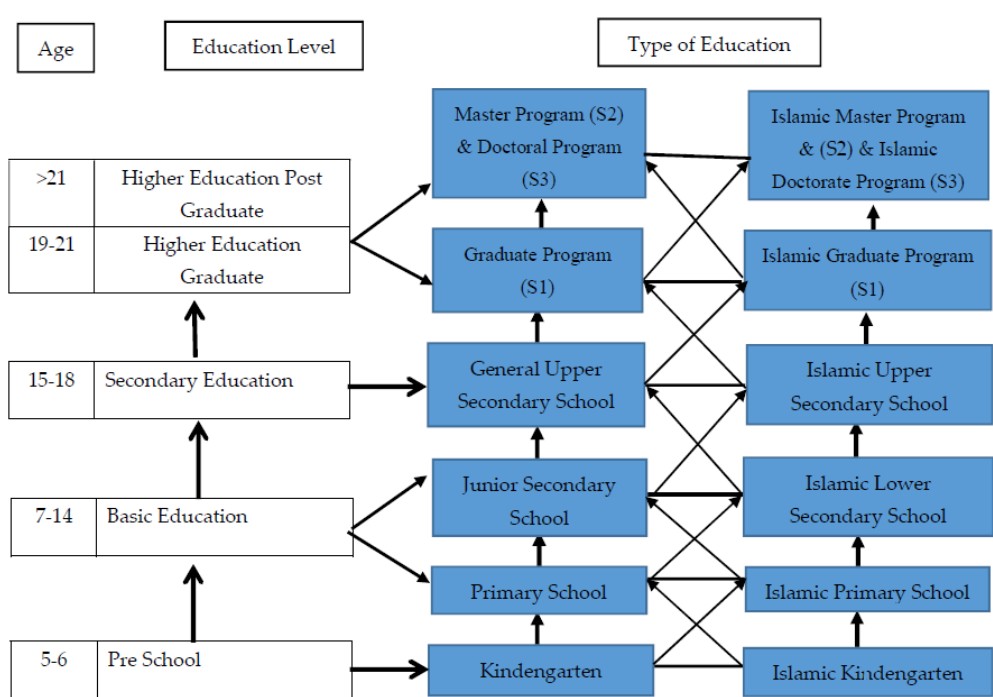

**Figure 1.** Education System in Indonesia, Law No. 20 2003. Source: (Kemendikbud 2016).

### 2.2. Children's Education

According to Primayana (2019), children's education is a conscious effort in inculcating character and morality. Awan (2013) believes that education is an important factor in developing human resources and creating opportunities for poverty alleviation. It can be the most effective means to improve the quality of life and the degree of community welfare (Allmendinger and Leibfried 2003). Education is the transmission of the cultural heritage and technological progress of a nation (Onyekuru and Ibegbunam 2013). To achieve this goal, it is necessary to have an active role from all elements of society, including families, to form quality future generations. Supporting this statement, Marin and Bocoş (2017); Rahmawati et al. (2017); Sapungan and Sapungan (2014) maintain that family involvement leaves positive impacts on the child's education process. Hamidun et al. (2019) further reveal that the synergy of parents and schools may lead to an increase in academic and non-academic achievement. Similarly, Sapungan and Sapungan (2014) find that the involvement of parents, especially mothers, in the process of children's education can improve their academic performance in which children are more capable of completing lessons and school assignments (Kwatubana and Makhalemele 2015).

Children whose mothers are actively involved in the process and the development of their education have a higher motivation not to give up easily when they do not understand a topic. They tend not to sleep in class because of a sense of responsibility (Sivertsen et al. 2015). Thus, children have greater self-confidence, a higher sense of responsibility to not disappoint their mothers and to behave positively, and they are more organized (Hornby and Lafaele 2011). Schools and parents alike can ensure the success of children in education. In contrast, Llamas and Tuazon (2016) and McNeal (2014) believe that a mother's involvement in child rearing may put pressure on a child because mothers tend to force children to perform well in every lesson, so children may feel distrusted by their mothers.

### 2.3. Working Mother

In this study, the employment status of mothers is limited to working and non-working mothers, but the discussion will focus on working mothers. Temitope (2015) defines working mothers as married women who have children. At the same time, they must work

to generate income, and they are responsible for house work. Similarly, Anafarta (2011) explains that working mothers are women who have children from the age of 0–18 years and become workers. Borjas (2013) further explains that the factors supporting working mothers include the level of wages, the comparison of market wages with reservation wages as well as the level of education, technology, leisure time, and skills (Borjas 2016). In favor of this statement, Cazzavillan and Olszewski (2011); Lugauer (2012); Greenlaw and Shapiro (2017); and Travaglini and Bellocchi (2018) believe that when women are highly skilled and have a high level of education, their interest in participating in the labor market increases. Besides, there can be other factors for women with children who decide to work. Duflo (2011), for example, maintains that the main reason for working mothers is the financial condition in which husbands are unable to work. In some cases, mothers also have the desire to improve the family's standard of living and to maintain a career (Heilman and Okimoto 2008), as well as a desire to improve social life and networking (Mendolia 2014). The other driving force for working mothers is their aspiration to have financial freedom and to not depend entirely on their husbands. Additionally, they want to fulfill their parents' needs and achieve self-actualization (Hasanah et al. 2017). More practically, they simply want to get family insurance benefits, pension benefits, and social security (Heinrich 2014). There are some who believe that working is a part of supporting national development (Duflo 2012).

These factors cause a shift in mothers' roles. Mothers must carry out dual roles to be a mother and a worker (OECD 2021). Several previous studies such as Barnett (2014); Ciciolla et al. (2017); Hildebrandt and McKenzie (2005); Ingstad and Hedlund (2017); Rizky et al. (2017) reported on the advantages and the disadvantages in the dual role of mothers due to the negative and positive effects this has caused in society and the family. A number of studies reveal the effects of working mothers on family welfare such as economic resilience (Casmini 2020); household consumption and positive health insurance 2017 (Rizky et al. 2017); and insurance ownership, availability of health care costs, the amount of available health information, and wider education (Laksono et al. 2021; Akresh et al. 2012). Mothers themselves tend to have a healthier soul, and they have lower rates of depression when working. They become role models for children because they can create a positive husband-and-wife relationship (Afiatin et al. 2016; Almani et al. 2012; Verma and Negi 2020). However, contrary to these findings, several previous studies have found negative effects on family resulting in vulnerability to family health and loss of child growth and development. This, in turn, has caused little time allocation for family and mothers being susceptible to stress due to fatigue (Botezat and Pfeiffer 2014; Kusumawardhani and Warda 2013; Lei et al. 2018; Rahmani et al. 2018; Osorio et al. 2014; Portner 2014; Rizky et al. 2017).

*2.4. Working Mother and Child Education's Relationship*

Working mothers cause separation between parents and children which leaves negative impacts (Rizky et al. 2019). Samman et al. (2016) confirm that working mothers result in reduced time in caring for children and this will influence life patterns in adulthood in areas such as education and employment. Bettinger et al. (2014) suggests the busyness of mothers due to participation in the labor market causes children to lose their character education base from their parents, which indirectly affects the first 9 years of schooling. In addition, children respond to the busyness of mothers by being arrogant, disrespectful, late for school, and even skipping school (Rizky et al. 2017). The findings of Portner (2014) explain that the absence of mothers in a child's learning process has negative effects, especially on girls, such as decreased achievement and absenteeism from school. Children whose mothers stopped working since the age of 4 with an average working hour of 30 h per week have lower academic achievement than children whose mothers stopped working when they were more than 10 years old (Dunifon et al. 2013). Furthermore, Hoque et al. (2017) more specifically found that one of the factors that influenced children's education and academic

grades was the mother's employment status such as being a teacher, a profession which enabled her children to perform much better than other children.

However, contrary to this statement, Purmini et al. (2016) reveal contrasting results suggesting that working mothers are very important in overcoming the problem of dropout rates in Indonesia due to the inability of parents to pay for their children's schooling. She elaborates that even if children are left behind when their mother works, children will not get excessive negative effects if they already have a strong bond of affection. Negative effects will also not happen if mothers can make quality time such as accompanying children when they study. In terms of positive impacts, working mothers can improve the family economy and support children's education. Sofa (2019) also claims that the positive impact of working mothers on children's welfare is greater than the risks. Among these is the achievement of children's education and increased opportunities to attend higher education (i.e., successfully enter college).

## 3. Methodology

### 3.1. Data

The data used in this study is panel data which is a combination of cross-section and time-series sourced from the three-wave Indonesia Family Live Survey (IFLS), IFLS-3 (2000), IFLS-4 (2007), and IFLS-5 (2014). In addition, data related to household characteristics and the employment status of mothers are available on the IFLS-3 (2000) basis. The Indonesia Family Live Survey (IFLS) is a longitudinal survey of a random sample of households involving questionnaires and anthropometric measurements. A longitudinal survey is a survey conducted on the same research sample over several survey periods. A longitudinal survey is also a survey that involves monitoring progress and/or changes in the research sample.

This study uses IFLS data in book 5 sections DLA (children's education) and section 3A DL (education). Book 5 was used for observation in 2007 by providing information related to respondents and household members under 15 years of age. Book 3A was used for observation in 2014 by providing information regarding respondents and household members aged 15 years or older (Strauss et al. 2016).

The unit of analysis is children aged 0–7 years observed in IFLS-3 (in 2000) and after children turn 7–14 years old the data is recorded in IFLS-4 (in 2007), and in IFLS-5 (in 2014) after they turn 17–21 years old. In other words, the data for children who became the unit of analysis in 2000 were traced back in 2007 and 2014. The number of samples in this study, after being processed according to the sample criteria and meeting the requirements were 3449 respondents, of which 1743 (about 50.54%) were boys and 1706 (about 49.46%) were girls. The mother's employment status was recorded in the IFLS-3 in 2000.

The dependent variable or outcomes in this study are children's education which is reflected by the child's school years. Information regarding children's school years was obtained from book 5, a section of DLA which provides educational information for children aged 15 years and under, and section 3 a section of DL which provides information on the education of children aged 15 years and over. Children between 7–14 years and 14–21 years were chosen due to data availability. Besides, these age groups were chosen to see the long-term effect of mothers' employment status on children's educational attainment based on school age and education level. Meanwhile, children aged 0–7 years were chosen because the first three years of life are the peak of children's rapid growth, both physically and cognitively. At this age, the brain is developing in complexity, and it is more sensitive to environmental influences (Papalia and Martorell 2014; Bogin 2015). In addition, at the age of 0–7 years, children go through two stages of cognitive development which are very important and affect children's education. In the first two years of life, children experience sensorimotor development, in which they begin to interact with instinctive reflexive actions until they have the ability to think about objects or event that are not closely related to their environment (Piaget 1991). Second, at the age of 2–7 years, children are in the pre-operational stage. Children begin to represent the world with words and pictures. This

indicates an increase in symbolic thinking which goes beyond the relationship of sensory information and physical actions. Interventions for child growth and development from an early age can improve the quality of children's growth and development which affects children's education and future (Stephiana and Wisana 2019).

The independent variable or variable of interest in this study is the mother's employment status. This is a dummy variable, namely 1 if the mother works and 0 if the mother does not work. The mother's employment status was recorded in the IFLS-3 in 2000. Mothers are those who had children aged 0–7 years in 2000. Therefore, this paper only focuses on the employment status of mothers in 2000, namely during the golden age of children or ages 0–7 years, to be able to see the short-term and long-term effects of working mothers on children's educational attainment. Meanwhile, the status of working mothers in 2007 and 2014 was not considered in this study. Control variables in this study are the character of the child, the character of the mother, the character of the father, and the character of the household. This is because children's education is not only influenced by one input at a time but by many inputs. Table 1 presents a summary of the statistics of all variables used in the paper.

**Table 1.** Description of Variables and Summary Statistics.

| Variable | Type/Description | Obs | Mean | SD |
|----------|------------------|-----|------|-----|
| **Outcomes** | | | | |
| children's education in 2007 | continuous, level/classification | 3449 | 4.772 | 2.108 |
| children's education in 2014 | continuous, level/classification | 3449 | 9.948 | 2.52 |
| **Variable of Interest** | | | | |
| Working Mother in 2000 | dummy (1: work participant) | 3449 | 0.55 | 0.498 |

Furthermore, this study has limitations, especially in data, so this study does not include the variables of parents' religious participation. Additionally, IFLS-3 (in 2000) did not include children's national exam scores as a variable. Meanwhile, IFLS-4 (in 2007) and IFLS-5 (in 2014) had too few samples.

### 3.2. Estimation Model

Analysis of the relationship between the dependent and the independent variables in this study is performed with the OLS estimation method and IV using a cross-section and estimating IFLS-4 (in 2007) and IFLS-5 (in 2014), separately.

This research follows Brooks-Gunn et al. (2010) and Magnuson et al. (2004) model with some renewal and combination included. It sees education as the result of investment in human capital which consists of the accumulation of capital, goods, and time, allowing it to affect children's outcomes in adulthood. This study also adopts the Kingdon and Teal (2002) and Hoque et al. (2017) model which examines the effect of the mother's profession on children's academic achievement and the effect of salary and student achievement. Finally, this study follows the model developed by Sofa (2019) which investigates the effect of a mother's decision to work on children's later outcomes. The difference in the model in this study is the existence of the mother's employment status as the independent variable and the child's education as the dependent variable which is measured by the length of the child's schooling. The measurement of children's school years is inspired by research (Purmini et al. 2016). The estimates are as follows:

$$E_{c\ 7–14} = \alpha + \beta MWS_{i2000} + X'_{i\ c,m,f,h2000}\gamma + \theta_R + \theta_T + \varepsilon_i \tag{1}$$

$$E_{c\ 14–21} = \alpha + \beta MWS_{i2000} + X'_{i\ c,m,f,h2000}\gamma + \theta_R + \theta_T + \varepsilon_i \tag{2}$$

The model explains that "$E_{c\ 7–14}$ is children's education at the age of 7–14 years" and $E_{c\ 14–21}$ is children's education at the age of 14–21 years. It measures educational

attainment. The working mother status in 200 is $MWS_{2000}$. It measures whether the mother was working in 2000. If the mother was working, the value is 1 and it is 0 if they did not work. The coefficient of interest variable to identify the effect of a mother's working status on children's education is $\beta$. Importantly, this model includes a vector of individual controls $X'_i$; the vector $X'_{i\,c,m,f,h2000}$ contains the other explanatory variables that absorb child (c), mother (m), father (f), and household (h) characteristic differences in 2000. Here, we control for children (c), including gender and age. Meanwhile, mother (m) in this case includes several characteristics such as education level, age, marital status, ability to read newspapers, religious participation, religious observance, and the use of Bahasa Indonesia at home in 2000, 2007, and 2014, following the dependent variable. The characteristic for father includes work status, age, education, religious observance, marital status, ability to read newspapers, religious participation, religious observance, and the use of Indonesian at home in 2000, 2007, and 2014, following the dependent variable. Meanwhile, household (h) characteristics include per capita expenditure, number of siblings, the status of living with parents, the religion of the head of the household, sanitation, quality of water sources, location (rural/city), use of cooking fuel, home ownership status, internet use, and distance to school in 2007 and in 2014, following the dependent variable. The fixed effect regional control variable in the form of a dummy island is $\theta_R$. The time fixed effect (year) is $\theta_T$ and $\varepsilon$ is the error term.

*3.3. Instrumental Variable (IV)*

One assumption that needs to be maintained to obtain an unbiased estimator from a regression equation with OLS is there should be no correlation between the independent variable and the residual, or $Cov(x, u) = 0$. Another possible case is when one or more explanatory variables have a relationship with the residual [$Cov(x_i\ \varepsilon_i)\ 0\ atau\ E(x_i\ \varepsilon_i)\ 0$]. When endogeneity occurs, it will make the estimator biased and inconsistent (Verbeek 2004). As in this study, it is suspected that there is a problem of endogeneity.

Endogeneity may occur due to omitted variables, measurement error, and simultaneity. In the case of this research, the correlation between the working mothers variable and errors is caused by omitted variables where certain factors correlate with working mothers and affect the length of education of children. However, these factors are not included in the model and are, therefore, detected as errors so that they disappear from the model. Second, there is simultaneity. That is the correlation between the dependent variable and the independent variable. This means that working mothers are a result of the need to fund children's education. Another possible cause is an increase in the length of children's education due to additional stimulus obtained from working mothers which eventually forces mothers to work.

When looking at the model in this study, using the mother variable in 2000 (earlier) and measuring it in 2007 and 2014, there was no strong simultaneity detected. However, endogeneity and the omitted variable may still be present, allowing the status of working mothers to be correlated with unobserved factors that affect children's education. These unobserved factors then correlate with the working mother variable (Verbeek 2004).

According to Chesher (2010) an equation containing endogeneity may create 3 problems if estimated in a single equation using the OLS method. First, the estimators are biased and inconsistent. Second, hypothesis testing becomes invalid. Third, forecasts are biased and inconsistent.

One alternative to overcome this problem is to use the instrumental variable method. Blundell et al. (2005) say that the instrumental variable (IV) can overcome the OLS bias resulting from the endogeneity of the independent variables. By using a suitable instrument that correlates with the causal variable of interest, IV corrects the endogeneity problem. Therefore, IV is not correlated with other determinants in the dependent variable. This variable has a clear effect on the independent variable.

Blundell et al. (2005) said that the instrumental variable IV can overcome the OLS bias that results from endogeneity of the explanatory variables. The IV corrects for the

endogeneity problem by using good instruments. This instrument is correlated with the causal variable of interest, the employment status of the mother, but it is uncorrelated with any other determinants of the dependent variable. This variable has a clear effect on the employment status of the mother in the first-stage of the regression, and the only reason for a relationship between a mother's decision to work and their employment status is from the first stage. According to Angrist and Pischke (2009), good instruments come from a combination of institutional knowledge and ideas about processes determining the variable of interest.

For this reason, IV provides a treatment effect on working mother status which is influenced by changes in exogenous regressors that meet the exclusion limit. The estimation of the treatment effect was carried out because the mother's employment status variable reacted to the instrument. That is, IV estimates the local average treatment effect (LATE) of the effect of the employment status of mothers on a subset of individuals. In this study, the LATE estimation used on the mother's employment status is influenced by decision-making.

This study uses the decision-making variable as an instrument variable (IV) to the mother's employment status. A mother's employment status may correlate with decision-making variables in the family. Mothers who decide to work may have the power within the family or are involved in decision-making. They may also be frequently involved in the decision-making process. Therefore, mothers who are involved in decision making or who decide their own work have a high probability of working tendencies (Unger and Crawford 1992). On the other hand, the decision making variable is a good exclusion restriction, since this variable is highly correlated with the probability of a mother's employment status, but there is no reason to believe these would be a correlation with any other variables that encourage a child's education. However, the decision-making variable is not correlated with other variables that could increase the length of a child's education. Therefore, the model used in Instrument Variable is as follow:

First-stage equation:

$$MWS_i = \beta_0 + \beta_1 \, decision \, making_i + X'_i\theta + \mu_i \tag{3}$$

Second-stage equation:

$$E_c = \alpha_0 + \alpha_1 \widehat{MWS}_i + \gamma X'_i + \sigma_d + \sigma_s + \varepsilon_i \tag{4}$$

In the first stage equation, MWS: dummy working mother is the dependent variable. This study uses decision making (Dm) as the instrument variable for working mothers in 2000. In addition, the $X_i$ vector contains other explanatory variables to describe the characteristics of individuals, and households (the same control variables as in the OLS estimation). This study also added a control variable in the form of a regional fixed effect used as a difference in the scope of different islands. Table 1 and Appendix A presents a summary of the statistics of all variables used in the paper.

## 4. Results

### 4.1. Short-Term Estimation Result

Oreopoulos (2006) believes that the IV method has been used frequently to evaluate a model because it is more accurate in predicting consistent treatment effect than OLS. The IV estimate must have a valid instrument to confirm the treatment effect, but it cannot control other factors to influence the desired result.

#### 4.1.1. Model First Stage Estimate of Decision Making in 2007

Table 2 shows the first stage estimates used for the basis of all IV estimates. The results showed that self-decision regarding work was positive and significant, statistically affecting the endogenous variable of working mother status. This instrument also passed the weakness instrument test, rejecting the null hypothesis of the weakness instrument. All

weak instrument *P* value tests are less than 0.01. Besides, the value of the P weakness of the instrument is less than 0.01 in the first model without including control variables, while the statistical model by including the control variable is significant at 0.001. Therefore, if mothers have the power to decide to work (at their own decision) the probability value of the opportunity to work increases by about 0.462 points (without a control variable) for the overall category of children 0–7 years. Meanwhile, the category of children aged 0–7 years by controlling for other variables increased by approximately 0.442 points compared with those without controlling for other variables.

**Table 2.** Model First Stage Estimation Decision Making 2007.

| Variables | (1) | (2) |
|---|---|---|
| Dependent variables: Mother working status in 2000 | Non-Control | With Control |
| | 0–7th | 0–7th |
| Dummy self-decision to work in 2000 | 0462 *** | 0.442 *** |
| | (0.0166) | (0.0215) |
| Control and Island Fixed Effects | No | Yes |
| F-stat of endogeneity test | 0.89 | 1.73 |
| *p*-value of endogeneity test | 0.35 | 0.19 |
| F-stat of weak Instrument | 770.12 | 422.56 |
| Observations | 3.173 | 1.835 |

Standard errors in parentheses. *** $p < 0.01$, Control variable of children characteristics ($I_c$) including gender, age, religious school. $I_m$: is the mother's characteristics in the form of education level, age, marital status, ability to read newspapers, religious participation, religious observance, praying five times a day, using Indonesian at home. $I_f'$ is the father's characteristics in the form of work status, age, education, religious observance, marital status, ability to read newspapers, religious participation, religious observance, pray five times a day, use Indonesian at home. $I_h'$: is household characteristics in the form of per capita expenditure, number of siblings, status of living with parents, religion of the head of the household, sanitation and quality of water sources, location (rural/city), use of cooking fuel, home ownership status, internet use and distance to school. $\theta_R$: fixed effect regional control variable in the form of dummy Island. $\theta_T$ is time fixed effect (year) and $\varepsilon$ is the error term.

### 4.1.2. OLS and IV Model Estimate of the Effect of Mother's Employment Status on Children's Education in 2007

Table 3 shows the results of the OLS estimation and the fourth estimate the effect of mother's employment status on children's education in 2007. OLS results show that working mothers in 2000 had a positive and significant influence on children's education in 2007. Without controlling for other factors, the result of the OLS estimate indicates that the length of children's education for mothers who worked in 2000 increased by 0.479 in 2007 for children aged between 0 and 7 years. Finally, the OLS estimation results by controlling for other factors (including control variables) reveal that children's education length for mothers who worked also rose by about 0.444 for the same age category (0–7). This means that working mothers in 2000 had a cumulative effect on children's education in 2007.

On the other hand, the result of the IV estimate shows a lower coefficient than OLS. Without controlling for other factors, the length of education for children whose mothers worked in 2000 increased by 0.337 in 2007 for children aged between 0 and 7 years. By controlling for other factors (including control variables), the estimation results illustrate that the duration of education for children whose mothers worked also inclined by 0.177 for the same age category.

The results show a positive bias because the IV estimate shows a lower coefficient. This study is in line with the research of Davies et al. (2017); Brookhart et al. (2006) which reveal that there is a ratio of differences in prevalence that is smaller than the power of the instrument so that the results of the instrumental variables tend to be positively biased and have a lower tendency to asymptotic bias. The study also follows the same trend as that of Angrist and Kruerger (1991) which shows the coefficient IV varies and it has a lower tendency due to analyzing a large sample. The results of this study, however, differ from the findings of Card (2001) in that the IV parameter is higher than OLS.

**Table 3.** OLS and IV Model Estimations of the influence of the working mother on children education attainment in 2007.

| Variables | (1) | (2) | (3) | (4) |
|---|---|---|---|---|
| Dependent Variable: children's education in 2007 | OLS 1 | OLS 2 | IV 1 | IV 2 |
| Dummy of Mothers' Working Status | 0.479 *** | 0.444 ** | 0.337 ** | 0.177 |
| | (0.0744) | (0.0997) | (0.168) | (0.2225) |
| Control and Island Fixed Effects | No | Yes | No | Yes |
| Observations | 3.173 | 1.835 | 3.173 | 1.835 |

Standard errors in parentheses *** $p < 0.01$, ** $p < 0.05$. Control variable of children characteristics ($I_c$) including gender, age, religious school. $I_m$: is the mother's characteristics in the form of education level, age, marital status, ability to read newspapers, religious participation, religious observance, praying five times a day, using Indonesian at home. $I_f'$ is the father's characteristics in the form of work status, age, education, religious observance, marital status, ability to read newspapers, religious participation, religious observance, pray five times a day, use Indonesian at home. $I_h'$: is household characteristics in the form of per capita expenditure, number of siblings, status of living with parents, religion of the head of the household, sanitation and quality of water sources, location (rural/city), use of cooking fuel, home ownership status, internet use and distance to school. $\theta_R$: fixed effect regional control variable in the form of dummy Island. $\theta_T$ is time fixed effect (year) and $\varepsilon$ is the error term.

### 4.1.3. First Stage Model Estimate of Decision Making in 2014

Next, Table 4 gives information about an estimate of the influence of working mothers in 2000 on children's education in 2014. This instrument also passed the weakness instrument test rejecting the null hypothesis of the weakness instrument. All weak instrument P value tests are less than 0.01. The value of the P weakness of the instrument is less than 0.01 in the first model, without including control variables, while the statistical model includes a significant control variable of 0.001.

**Table 4.** Model First Stage Estimation Decision Making 2014.

| Variables | (1) | (2) |
|---|---|---|
| Dependent variables: Mother working status in 2000 | | |
| Dummy self-decisions to work in 2000 | 0.462 *** | 0.427 *** |
| | (0.0166) | (0.0282) |
| Control and Island Fixed Effects | No | Yes |
| F-stat of endogeneity test | 1.62 | 0.35 |
| *p*-value of endogeneity test | 0.2 | 0.55 |
| F-stat of weak Instrument | 770.12 | 229.03 |
| Observations | 3.173 | 1.142 |

Standard errors in parentheses *** $p < 0.01$, Control variable of children characteristics ($I_c$) including gender, age, religious school. $I_m$: is the mother's characteristics in the form of education level, age, marital status, ability to read newspapers, religious participation, religious observance, praying five times a day, using Indonesian at home. $I_f'$ is the father's characteristics in the form of work status, age, education, religious observance, marital status, ability to read newspapers, religious participation, religious observance, pray five times a day, use Indonesian at home. $I_h'$: is household characteristics in the form of per capita expenditure, number of siblings, status of living with parents, religion of the head of the household, sanitation and quality of water sources, location (rural/city), use of cooking fuel, home ownership status, internet use and distance to school. $\theta_R$: fixed effect regional control variable in the form of dummy Island. $\theta_T$ is time fixed effect (year) and $\varepsilon$ is the error term.

In 2014, if the mother has the power to decide to work (at her own decision), the probability value of the opportunity to work increases by approximately 0.462 points (without a control variable) for the category of children aged 0–7 years. Meanwhile, the category of children aged 0–7 years by controlling for other variables increased by approximately 0.427 points.

### 4.1.4. IV Model Estimate of the Effect of Mother's Employment Status on Children's Educational Attainment in 2014

Table 5 shows the results of the OLS estimation and IV estimation of the effect of mother's employment status on children's education in 2014. The OLS results show that working mothers in 2000 had a positive and significant influence on children's education in 2014. Without controlling for other factors, the result of the OLS estimate indicates that the length of education for children whose mothers worked in 2000 increased by 0.344 in 2014 for children aged between 0 and 7 years. Finally, the OLS estimation results by controlling for other factors (including control variables) reveal that children's education length whose mothers work also rose by approximately 0.244 for the same age category (0–7). This means that working mothers in 2000 had a cumulative effect on children's education in 2014. The results of the IV estimate show that the education of children whose mothers worked in 2000 increased by 0.575 points without controlling for other variables. However, the estimation results when controlling for other factors (including control variables), show that the education of children whose mothers worked increases by about 0.399. The result of the IV estimate shows a higher coefficient than OLS. This finding is in line with Card's (2001) finding which reveals that the IV parameter is higher than OLS because IV estimates the local average treatment effect (LATE) of the working mother status impact on a subset of individuals.

**Table 5.** IV Model Estimate and OLS of the effect of mother's employment status on children's educational attainment in 2014.

| Variables | OLS (1) | OLS (2) | IV (1) | IV (2) |
|---|---|---|---|---|
| Dependent Variable: children's education in 2014 | | | | |
| Dummy of Working Status | 0.344 *** | 0.244 ** | 0.575 *** | 0.399 *** |
| | (0.0889) | (0.117) | (0.202) | (0.399) |
| Control and Island Fixed Effects | No | Yes | No | Yes |
| Observations | 3.173 | 1.142 | 3.173 | 1.142 |

Standard errors in parentheses *** $p < 0.01$, ** $p < 0.05$, Control variable of children characteristics ($I_c$) including gender, age, religious school. $I_m$: is the mother's characteristics in the form of education level, age, marital status, ability to read newspapers, religious participation, religious observance, praying five times a day, using Indonesian at home. $I_f{}'$ is the father's characteristics in the form of work status, age, education, religious observance, marital status, ability to read newspapers, religious participation, religious observance, pray five times a day, use Indonesian at home. $I_h{}'$: is household characteristics in the form of per capita expenditure, number of siblings, status of living with parents, religion of the head of the household, sanitation and quality of water sources, location (rural/city), use of cooking fuel, home ownership status, internet use and distance to school. $\theta_R$: fixed effect regional control variable in the form of dummy Island. $\theta_T$ is time fixed effect (year) and $\varepsilon$ is the error term.

## 5. Discussion

Education for children is an effort to build a foundation in children to shape their future. Providing proper education for children is very crucial to support their growth and development in all fields. Through education, children will learn to grow their cognitive and social abilities to prepare themselves to enter a higher level of education. Family support as the smallest institution both materially and non-materially is a factor in the success of children's education. This study suggests that the mother's employment status supports the child's education both materially and non-materially. Material support includes the availability of children's education costs and cash transfers to fulfill daily consumption needs. Meanwhile, non-material support resulting from working mothers is in the form of an increase in the number of years of schooling and socio-economic status in the future. This finding is in line with that of Ferriss (2002) which shows that working mothers positively correlated with family welfare, providing both material and non-material welfare.

Mother's employment status indeed has a significant positive effect on children's education in the short-term in 2007 and the long-term in 2014. This finding is in line with previous research conducted by Purmini et al. (2016). She suggests that working mothers

have a positive effect on children's education. This is evident especially in Indonesia where the high dropout rate is mainly caused by parents' inability to afford children's school fees. In line with Abrar ul Haq et al. (2015); Lin and Lv (2017), the higher the family income, the higher the education of the child. This finding also supports the findings of Jayawardana et al. (2021) related to increasing family income, receiving government benefits in the form of goods, scholarships for school goods, reducing child labor rates, and having a positive impact on children's education. In addition, working mothers also have a positive effect on the level of household consumption where the higher household expenditure per month, the higher the education of children (Nurrika et al. 2020). However, Hoque et al. (2017) suggest that family income is not the sole contributing factor. They find that one of the factors that influence children's education and academic scores at school is the type of job that a mother has, especially when a mother works in the teaching field. Because children whose mothers work as teachers will not have excessive negative effects, they will support the mindset and educational process, creating a strong bond of affection. This study also provides support for Sofa's (2019) study regarding the low risk of working mothers and the high benefit it has on long-term education of children, namely at the age of 21 when they begin higher education (successfully enter college).

## 6. Conclusions

Children's education is the most crucial problem in developing countries like Indonesia. The high dropout rate and the low quality of education in Indonesia are due to several reasons. One of them is parents' inability to afford school fees for children. Besides, the level of family income and type of work is considered to have an important role in the level of education of children. This study tries to find the effect of working mothers on children's education level. The results showed that the mother's employment status had a significant positive effect on the increase in children's education levels in the short-term in 2007 and the long-term in 2014. In addition, participation in decision-making for mothers will increase the opportunities for mothers to work.

This study not only adds to the literature related to working mothers but also provides policy implications regarding the education of children and working mothers. Children's education is a long-term investment for the family and the country. Education plays an important role in the development and economic growth of the country. Therefore, every family must be able to provide good facilities for children's education. It is also important to note that the presence of mothers plays an important role in the education process of children. Thus, mothers who are working must also prepare a strong self-border and family-facilitation concept in a family to be able to minimize conflict. Mothers as the main facilitator of education must have a strong synergy with the school and the teachers. They need to be selective in choosing schools for their children because the role of teachers and environmental conditions have a positive effect on the child's education process. In addition, positive synergy with the company where the mother works is highly required. The company's policy towards working mothers indirectly supports children's education, such as policies on working hours, regulations on worship times during working hours, providing a child-friendly work environment, as well as the availability of quality daycare or early childhood education in the company area or workplace. Suggestions related to findings on government policies are related to expansion and equitable access to children's education, equitable distribution of children's education facilities, and equitable distribution of quality teachers, so that the quality of children's education systems and facilities are not only concentrated in urban areas. In addition, it is also important to improve the quality of religious-based schools under the auspices of the ministry of religion such as MI, MTs, Madrasah Aliyah, and Islamic Boarding School levels as parents' interest in sending their children to religious-based schools is considered very low.

**Author Contributions:** Conceptualization, S.N.A., S.S. and E.S.; methodology, E.S. and S.S.; validation, S.S., E.S. and S.N.A.; formal analysis, S.N.A.; investigation, S.N.A. and E.S., resources, S.S. and E.S., data curation, E.S. and S.N.A.; writing, S.N.A.; preparation, S.S. and S.N.A.; writing—review

and editing, E.S. and S.N.A.; supervision, S.S.; project administration, S.N.A.; funding acquisition, S.N.A. All authors have read and agreed to the published version of the manuscript.

**Funding:** This research received no external funding.

**Institutional Review Board Statement:** Not applicable.

**Informed Consent Statement:** Informed consent was obtained from all subjects involved in the study.

**Data Availability Statement:** Data supporting reported results can be found by asking directly of the first author.

**Conflicts of Interest:** The authors declare no conflict of interest.

## Appendix A

| Variable | Type/Description | Obs | Mean | SD |
|---|---|---|---|---|
| **Outcomes** | | | | |
| children's education in 2007 | continuous, level/classification | 3449 | 4.772 | 2.108 |
| children's education in 2014 | continuous, level/classification | 3449 | 9.948 | 2.52 |
| **Variable of Interest** | | | | |
| Working Mother in 2000 | dummy (1: work participant) | 3449 | 0.55 | 0.498 |
| **Children's Characteristics (X1)** | | | | |
| Child's Age in 2000 | continuous | 3449 | 3.366 | 2.059 |
| Child's Age in 2007 | continuous | 3449 | 10.75 | 2.006 |
| Child's Age in 2014 | continuous | 3449 | 17.664 | 2.003 |
| Child is Male 2000 | dummy (1: male) | 3449 | 0.505 | 0.5 |
| Child is Male 2007 | dummy (1: male) | 3449 | 0.505 | 0.5 |
| Child is Male 2014 | dummy (1: male) | 3449 | 0.506 | 0.5 |
| Child attended school in 2007 | dummy (1: In school) | 3449 | 0.964 | 0.186 |
| Child attended school in 2014 | dummy (1: In school) | 3449 | 0.595 | 0.491 |
| Kids Islamic School 2007 | dummy (1: Islamic school) | 3400 | 0.107 | 0.31 |
| Kids Islamic School 2014 | dummy (1: Islamic school) | 3428 | 0.12 | 0.326 |
| **Mother's Characteristics (X2)** | | | | |
| **2000** | | | | |
| Mother Working Status | dummy (1: work participant) | 3449 | 0.55 | 0.498 |
| Mother working hours | continuous | 1897 | 38.753 | 24.926 |
| Mother's ages | continuous | 3449 | 30.646 | 6.618 |
| Mother's education | continuous | 3448 | 7.27 | 4.063 |
| Mother's education level | continuous | 3448 | 2.698 | 1.051 |
| Mother's marital status | categorical | 3449 | 1.068 | 0.398 |
| Work self dec mother | dummy (1: self decision) | 3173 | 0.649 | 0.477 |
| Religious mother | dummy (1: religious) | 0 | 0 | 0 |
| Mother pray five times | dummy (1: pray five times) | 0 | 0 | 0 |
| Mother read news paper | dummy (1: can read the newspaper | 3349 | 0.822 | 0.323 |
| mother uses Indonesian | dummy (1: mother can speak Indonesian) | 3349 | 0.385 | 0.487 |
| Mother's religious participation | dummy (1: religi participant) | 0 | 0 | 0 |

| Variable | Type/Description | Obs | Mean | SD |
|---|---|---|---|---|
| **2007** | | | | |
| Mother Working Status | dummy (1: work participant) | 3220 | 0.652 | 0.476 |
| Mother working hours | continuous | 2124 | 39.742 | 34.786 |
| Mother's ages | continuous | 3220 | 38.018 | 6.559 |
| Mother's education | continuous | 3220 | 7.37 | 4.177 |
| Mother's education level | continuous | 3220 | 2.733 | 1.076 |
| Mother's marital status | categorical | 3220 | 1.132 | 0.573 |
| Work self dec mother | dummy (1: self decision) | 2856 | 0.813 | 0.39 |
| Religious mother | dummy (1: religious) | 3220 | 0.994 | 0.077 |
| Mothers pray five times | dummy (1: pray five times) | 2782 | 0.871 | 0.335 |
| Mothers read newspaper | dummy (1: can read the newspaper | 3220 | 0.891 | 0.321 |
| mother uses Indonesian | dummy (1: mother can speak Indonesian) | 3220 | 0.353 | 0.478 |
| Mothers religious participation | dummy (1: religi participant) | 2915 | 0.679 | 0.467 |
| **2014** | | | | |
| Mother Working Status | dummy (1: work participant) | 2559 | 0.72 | 0.49 |
| Mother working hours | continuous | 1887 | 39.289 | 26.057 |
| Mother's ages | continuous | 2599 | 44.705 | 6.609 |
| Mother's education | continuous | 2599 | 7.344 | 4.246 |
| Mother's education level | continuous | 2599 | 2.734 | 1.076 |
| Mother's marital status | categorical | 2599 | 1.279 | 0.827 |
| Work self dec mother | dummy (1: self decision) | 2146 | 0.684 | 0.465 |
| Religious mother | dummy (1: religious) | 2541 | 0.983 | 0.128 |
| Mothers pray five times | dummy (1: pray five times) | 2274 | 0.893 | 0.31 |
| Mothers read newspaper | dummy (1: can read the newspaper | 2599 | 0.868 | 0.339 |
| mother uses Indonesian | dummy (1: mother can speak Indonesian) | 2599 | 0.375 | 0.484 |
| Mothers religious participation | dummy (1: religi participant) | 2254 | 0.723 | 0.448 |
| **Father's Characteristics (X3)** | | | | |
| **2000** | | | | |
| Father's Working Status | dummy (1: work participant) | 2942 | 0.984 | 0.125 |
| Father's working hours | continuous | 2917 | 47.093 | 26.286 |
| Father's ages | continuous | 2972 | 35.515 | 7.967 |
| Father's education | continuous | 2969 | 7.916 | 4.343 |
| Father's education level | continuous | 2969 | 2.902 | 1.129 |
| Father's marital status | categorical | 2972 | 1 | 0 |
| Work self dec Father's | dummy (1: self decision) | 2970 | 0.973 | 0.161 |
| Religious Fathers | dummy (1: religious) | 0 | 0 | 0 |
| Father pray five times | dummy (1: pray five times) | 0 | 0 | 0 |
| Father read news paper | dummy (1: can read the newspaper | 2972 | 0.916 | 0.277 |
| Father uses Indonesian | dummy (1: fathers can speak Indonesian) | 3449 | 0.882 | 0.323 |
| Father religious participation | dummy (1: religi participant) | 0 | 0 | 0 |

| Variable | Type/Description | Obs | Mean | SD |
|---|---|---|---|---|
| **2007** | | | | |
| Father's Working Status | dummy (1: work participant) | 2761 | 0.97 | 0.171 |
| Father's working hours | continuous | 2678 | 45.118 | 21.544 |
| Father's ages | continuous | 2761 | 42.589 | 7.552 |
| Father's education | continuous | 2761 | 8.083 | 4.434 |
| Father's education level | continuous | 2761 | 2.954 | 1.142 |
| Father's marital status | categorical | 2761 | 1.022 | 0.246 |
| Work self dec Father's | dummy (1: self decision) | 2699 | 0.983 | 0.128 |
| Religious Fathers | dummy (1: religious) | 2760 | 0.993 | 0.083 |
| Father pray five times | dummy (1: pray five times) | 2180 | 0.846 | 0.361 |
| Father read news paper | dummy (1: can read the newspaper | 2761 | 0.921 | 0.27 |
| Father uses Indonesian | dummy (1: Father can speak Indonesian) | 2761 | 0.356 | 0.479 |
| Father religious participation | dummy (1: religi participant) | 2545 | 0.679 | 0.467 |
| **2014** | | | | |
| Father's Working Status | dummy (1: work participant) | 2183 | 0.947 | 0.223 |
| Father's working hours | continuous | 2054 | 44.365 | 20.785 |
| Father's ages | continuous | 2183 | 49.144 | 7.497 |
| Father's education | continuous | 2182 | 8.087 | 4.49 |
| Father's education level | continuous | 2182 | 2.966 | 1.139 |
| Father's marital status | categorical | 2183 | 1.062 | 0.411 |
| Work self dec Father | dummy (1: self decision) | 1965 | 0.96 | 0.196 |
| Religious Fathers | dummy (1: religious) | 2043 | 0.996 | 0.066 |
| Father pray five times | dummy (1: pray five times) | 1712 | 0.846 | 0.361 |
| Father read news paper | dummy (1: can read the newspaper | 2183 | 0.909 | 0.288 |
| Father uses Indonesian | dummy (1: fathers can speak Indonesian) | 2183 | 0.389 | 0.488 |
| Father religious participation | dummy (1: religi participant) | 1806 | 0.738 | 0.44 |
| **Household Characteristics (X4)** | | | | |
| **2000** | | | | |
| Per Capita Consumption expenditure | continuous/counts | 14,591 | 545,774 | 808,075 |
| Inpce | log per capita consumption expenditure | 14,591 | 12.684 | 0.993 |
| Head of household religion | dummy (1: islam) | 14,941 | 0.899 | 0.301 |
| Urban | dummy (1: city) | 14,943 | 0.519 | 0.5 |
| Island | dummy (1: java) | 14,943 | 0.529 | 0.499 |
| Province | categorical | 14,943 | 34.62 | 15.909 |
| Island | categorical | 14,943 | 2.152 | 1.011 |
| Live with mother | dummy (1: live together) | 0 | 0 | 0 |
| Live with father | dummy (1: live together) | 0 | 0 | 0 |
| Own House | dummy (1: own house) | 3216 | 0.839 | 0.368 |
| HH Has good Toilet | categorical | 3449 | 2.245 | 1.281 |
| HH Has good Clean Drink Water | dummy (1: house with a clean drink water) | 3449 | 0.874 | 0.332 |

| Variable | Type/Description | Obs | Mean | SD |
|---|---|---|---|---|
| HH Has clean stove | dummy (1: house with a good stove) | 3449 | 0.105 | 0.307 |
| HH time to school | continuous | 0 | 0 | 0 |
| HH Use Internet | dummy (1: use internet) | 0 | 0 | 0 |
| Number Sibling | | 0 | 0 | 0 |
| **2007** | | | | |
| Per Capita Consumption expenditure | continuous/counts | 3390 | 456,068 | 371,986 |
| Inpce | log per capita consumption expenditure | 3390 | 12.812 | 0.634 |
| Head of household religion | dummy (1: islam) | 3448 | 0.899 | 0.302 |
| Urban | dummy (1: city) | 3449 | 0.511 | 0.5 |
| Island | dummy (1: java) | 3449 | 0.514 | 0.5 |
| Province | categorical | 3449 | 34.758 | 15.781 |
| Live with mother | dummy (1: live together) | 2933 | 0.959 | 0.199 |
| Live with father | dummy (1: live together) | 2794 | 0.906 | 0.293 |
| Own House | dummy (1: own house) | 3279 | 0.844 | 0.362 |
| HH Has good Toilet | categorical | 3449 | 1.815 | 1.185 |
| HH Has good Clean Drink Water | dummy (1: house with clean drink water) | 3449 | 0.879 | 0.326 |
| HH Has clean stove | dummy (1: house with a good stove) | 3449 | 0.186 | 0.389 |
| HH time to school | continuous | 3419 | 0.203 | 0.192 |
| HH Use Internet | dummy (1: use internet) | 0 | 0 | 0 |
| Number Sibling | count | 3375 | 2.23 | 1.609 |
| **2014** | | | | |
| Per Capita Consumption expenditure | continuous/counts | 3238 | 1,197,474 | 1,159,774 |
| Inpce | log per capita consumption expenditure | 3238 | 13.733 | 0.689 |
| Head of household religion | dummy (1: islam) | 3449 | 0.904 | 0.295 |
| Urban | dummy (1: city) | 3449 | 0.649 | 0.477 |
| Island | dummy (1: java) | 3449 | 0.52 | 0.5 |
| Province | categorical | 3449 | 34.829 | 15.644 |
| Live with mother | dummy (1: live together) | 2447 | 0.926 | 0.262 |
| Live with father | dummy (1: live together) | 2356 | 0.831 | 0.375 |
| Own House | dummy (1: own house) | 2974 | 0.815 | 0.388 |
| HH Has good Toilet | categorical | 3449 | 1.535 | 0.97 |
| HH Has good Clean Drink Water | dummy (1: house with clean drink water) | 3449 | 0.905 | 0.293 |
| HH Has clean stove | dummy (1: house with a good stove) | 3449 | 0.758 | 0.429 |
| HH time to school | continuous | 3287 | 0.347 | 0.328 |
| HH Use Internet | dummy (1: use internet) | 3440 | 0.843 | 0.364 |
| Number Sibling | count | 2834 | 2.347 | 1.553 |

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
