# Peer review of "The Effect of Working Mother Status on Children’s Education Attainment: Evidence from Longitudinal Data"

_economies, doi:10.3390/economies10020054_

Round 1

Reviewer 1 Report

I do not make any significant suggestions. The article is well done.

Extensive literature review shows well scientific gap, which the authors discuss and solve in the next chapters.

The authors comprehensively analyze the issue and use the correct methods. The conclusions are orginal and based on the presented calculations.

From my point of view, the issue of working mothers and their children's education is very interesting and the conclusions are useful on many levels.

I fully recommend the article for publication.

Author Response

We as writers are very grateful for the suggestions from all reviewers. Your suggestions and criticisms are very valuable for us so that in the future we will be better at writing articles.

Sincerely

Reviewer 2 Report

This study tries to determine the extent to which mothers’ employment status affects children's education in the short and long term. (line 95)

I find the topic very interesting and important for shaping proper policies in order to facilitate development of society.

There is already a certain inconsistency in the abstract:

This study uses data on children from three waves in the Indonesia Family Life Survey (IFLS), IFLS-3 (in 2000) aged 0-7 years, IFLS-4 (in 2007) aged over 7-14 years, and IFLS-5 (in 2004) aged over 14-21 years.” At this point it is not clear why 2nd wave is not treated as a third, and 3dr is not treated as a second. But everything becomes clear while looking at part “3.1 Data”. There is simple mistake in the abstract (2004 should be 2014).

Still in abstract there is an information:

This study also used a cross-sectional design which estimated wave 4 (IFLS-2007) and wave 5 (IFLS-4) separately.

Why Authors introduce inconsistency at the very beginning naming own data “IFLS-4” vs “IFLS-2007”? Are these different databases?? I suppose not. That’s why my suggestion is once used abbreviation should be used consistently.

(Lines 240-241): “(…) using a cross-section and estimating wave 4 (IFLS-2007) and wave 5 (IFLS-4) separately.” This is exactly the same problem I mentioned above – inconsistency in used abbreviations. I suggest Authors should keep themselves descriptions like this: IFLS-2000, IFLS-2007, IFLS-2014. It is clear, informative and does not confuse the number of wave with the counting of databases used for calculations (e.g. wave 5 (IFLS-4)).

Talking about methodology, I wonder why there are such age brackets like described in the manuscript. Why not: 0-7; 8-14; 15-21?

Presser (2000) even suggests that husband and wife are six times more at risk for divorce when mothers work. This is because working mothers have little time allocation for interaction with children and families (Singh 2018). (line 70-72)

I would add: this is also because a mother who starts to work and gets her own income source makes her more independent from husband. And it is easier for her to decide about divorce if her husband is just bad for her. It is easy to find examples of analysis about that phenomenon.

(Lines 235-237) Author probably should explain what kind of “inputs” they mean here or should add information that Table 1 explains everything.

Table 1: I do not know the system of education in Indonesia and wonder if there is a rationale for dummy “Child attended school in 2000=1” if there are only kids of age 0-7. Because the role of such information is influenced mainly by the age structure of sample, not by other (key) variables. I wander what valuable information introduces “0” here in cases of kids of pre-school age.

In general I am not sure if children’s age should not be divided different way here. I mean to cut pre-school kids from the calculations leaving only for instance kids between 6 (7) and 21 y.o..

At the same time I take into account explanation in lines 267-270.

In addition, participation in decision-making for mothers will increase the opportunities for mothers to work. However, this is not the case in families where husbands are the sole decision-maker. (Line 483-485) This part seem to be a comment for results of different research with different aim. It is to short to enrich conclusions for this work and additionally introduce another thread which needs more investigation.

And my general impression: abstract and first part of the text seem to announce that main tool used to calculate will be OLS with only additional help of “IV”. And the next parts with results of IV seem to show IV as main tool and OLS’s results are only background (not much described with numbers).

Notes of little importance:

Line 62: 17 years-70 years should be 17-70 years.

Lines 509-511: this passage should be moved to the part describing the data sets used. For sure it does not work as a very last part of whole manuscript.

Author Response

Dear Reviewer

We as writers are very grateful for the suggestions from all reviewers. Your suggestions and criticisms are very valuable for us so that in the future we will be better at writing articles.

Attached our response

Sincerely

Reviewer 3 Report

The paper is well written and the authors demonstrate to know the literature very well. The paper misses though important explanations and has some flaws, especially in the methodological part. I provide my main concerns in the document attached.

Referee Report on “The Effect of Working Mother
Status on Children’s Education Attainment: Evidence from Longitudinal Data”
Summary
The paper examines the causal impact of working mothers on children education. Using a longitudinal Survey in Indonesia which was implemented in three waves, it shows that there seems to be an association between working mother status and children education.
Using a decision-making variable as an instrument for working mother status, it also shows that the relationship between working mother status and children education is causal.
Overall assessment
The paper is well written and the authors demonstrate to know the literature very well.
Although I am not an expert in the field, the paper seems innovative but misses important explanations and details, especially in the methodological part. The paper is also not very well polished. I list below my main concerns.
Comments
Since the authors apply their analysis to a very specific country, it might be worth writing a section on the institutional background. For example, the authors say that children education is reflected by children’ school years. My question is: when does a child start her/his education? When she/he is 5,6, or 7 years old?
The survey design is also not explained. I give an example about the unit of analysis (also related to the previous question). The authors say that in 2000, we can observe have children aged 0-7. Then in 2007, if the authors observe the same children, how her/his years of education are computed?
From page 5 to 10, the authors provide a full list of variables. It might be better to put the less relevant ones in the Appendix because the reader might get confused. For example, the data shows the working status of the mother in 2000,2007 and 2014. Then in equation 1 and 2 the authors only refer to the working status in 2000. So it is better to only put the working status in 2000 in the summary stats and put all summary stats
not relevant to the analysis in the Appendix. Related to the working status of the mother, I have a more methodological question.
If the authors only refer to 2000, what happens if a mother stops working across the 1 different waves of the survey? The authors do not discuss this problem at all in their analysis.
Section 4.1 provides equation 1 and 2 but some variables are not explained. What is y1, y2, y3? They are indicator variables but I think the authors only need to explain Is (as they do) without including y1, y2, y3. Also in the equations the authors forgot β, that I think it is the coefficient for MW S2000. The equation and subscripts can also be defined much better. What is Θt? I guess it is a control for the year of the survey but I had to guess it.
In my opinion, it is better to provide the OLS results at the end of Section 4.1 and then in Section 4.2 (as the authors do) discuss the endogeneity issues.
The endogeneity issues are well discussed in Section 4.2 but the section misses a clear explanation of the decision making variable. Is this a variable available in the survey? Which is the question asked in the survey that allow to construct this measure? Also, I do not believe that this variable satisfies the exclusion restriction. I think that the decision-making process in the family can also directly affect children education and not just through the working status of the mother. But I might be convinced otherwise if
there is some literature on this. The authors can also refer to other instruments used in the literature that analyze similar problems. Although not related to this case, the literature sometimes uses birth date as an instrument for education.
Is the dummy indicating the self-decision to work related to the year 2000? Or is it related to all years of the survey? The authors do not say that.
More formal (but still very important) comments
Results in Tables 2,3,4,5 are presented without standard errors, although the authors refer to standard errors in the notes of the tables. It is important to include them in general, even more here where some of the relationships are not statistically different from 0.
Table 2 and 4 should also report the dependent variable of the first stage. The reader can only guess it from the equation at the end of page 11.

Author Response

Dear Reviewer 3

We as writers are very grateful for the suggestions from all reviewers. Your suggestions and criticisms are very valuable for us so that in the future we will be better at writing articles.

Attached our response           

Sincerely

Reviewer 4 Report

My comments are in the included pdf file (Revision.pdf)

Author Response

(The authors gave the same response as above.)

Round 2

Reviewer 3 Report

Dear authors,

thank you for having addressed my comments. I am satisfied with your answers.

Author Response

Dear Reviewer

We as writers are very grateful for the suggestions from all reviewers. Your suggestions and criticisms are very valuable for us so that in the future we will be better at writing articles.

Yours sincerely,

Siti Nur Azizah

Reviewer 4 Report

p.13

Religious mother dummy (1: religius) 3,220 0.994 0.77

instead of 0.77 should be 0.077

'religius' is not in English but I think Indonesian

Author Response

Dear Reviewer

We as writers are very grateful for the suggestions from all reviewers. Your suggestions and criticisms are very valuable for us so that in the future we will be better at writing articles.

 sincerely,
